# IMITATION LEARNING FROM PIXEL OBSERVATIONS FOR CONTINUOUS CONTROL

## ABSTRACT

We study imitation learning using only visual observations for controlling dynamical systems with continuous states and actions. This setting is attractive due to the large amount of video data available from which agents could learn from. However, it is challenging due to $i)$ not observing the actions and $ii)$ the high-dimensional visual space. In this setting, we explore recipes for imitation learning based on adversarial learning and optimal transport. A key feature of our methods is to use representations from the RL encoder to compute imitation rewards. These recipes enable us to scale these methods to attain expert-level performance on visual continuous control tasks in the DeepMind control suite. We investigate the tradeoffs of these approaches and present a comprehensive evaluation of the key design choices. To encourage reproducible research in this area, we provide an easy-to-use implementation for benchmarking visual imitation learning, including our methods[1].

## 1 INTRODUCTION

Learning continuous control policies directly from pixel observations is an important problem due to its potential impact on fields like robotics, autonomous driving and video games. These domains have rich resources of data available of humans performing expert-level demonstrations that our software agents do not leverage as they are often trained from scratch without any knowledge of how humans think about these problems. However, using unlabeled video data is challenging as it $i)$ requires distilling a representation of the world into the policy of an agent, and $ii)$ we do not know the underlying actions and reasoning process of the expert. This renders common algorithms like canonical behavioral cloning (Pomerleau, 1988; 1991) useless in the no-action setting.

Recently the community has advanced our understanding in learning visual representations and learning to imitate demonstrations provided as proprioceptive states. Visual representation learning has been crucial in recent advancements for sample-efficient reinforcement learning (RL) directly from pixels in continuous spaces, *e.g.* with reconstruction (Finn et al., 2016; Yarats et al., 2019), contrastive learning (Srinivas et al., 2020; Stooke et al., 2020), unsupervised pre-training (Liu & Abbeel, 2021; Yarats et al., 2021b; Seo et al., 2021), world models (Hafner et al., 2018; 2019; 2020) and data augmentation (Yarats et al., 2021c; Raileanu et al., 2020; Laskin et al., 2020).

These approaches require known reward signals from the environment, which are not always available or well-defined. When expert demonstrations are available, imitation learning (IL) and inverse RL (IRL) methods overcome the issue of not having a reward signal and seek to recover the expert agent (Ng & Russell, 2000). These methods are very effective when low-dimensional proprioceptive states and actions are available and typically consist in learning based on the mismatch between the expert and agent's state(-action) distributions.

In this paper, we combine the budding areas of model-free image-based reinforcement learning and state-action imitation learning to control non-trivial continuous dynamical systems from pixel demonstrations. We extend two leading proprioceptive-state approaches to comparing the pixel trajectories of the learner and expert – see fig. 1 for an illustration. The first approach, *pixel sinkhorn imitation learning*, P-SIL, extends optimal transport approaches for imitation learning to the image

---

[1]Code at https://anonymous.4open.science/r/ImitateFromPixelsICLR-B0EF

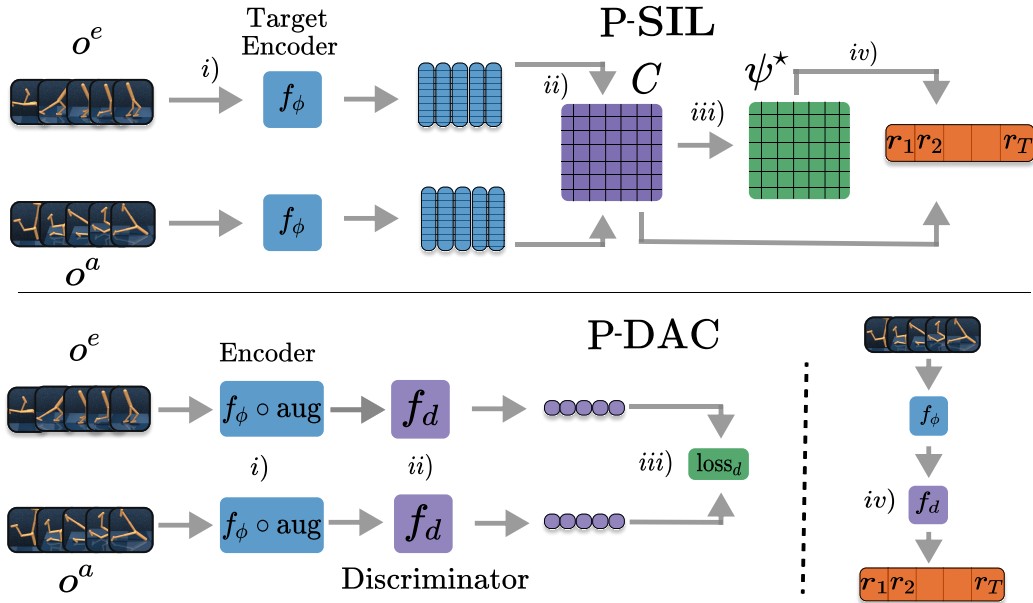

Figure 1: Summary of our proposed methods P-SIL and P-DAC. Top: P-SIL, $i)$ encodes the agent $o^a$ and expert $o^e$ trajectories into a latent space, $ii)$ computes a cost matrix $C$ and $iii)$ transport map $\pi^\star$ between these to $iv)$ produce imitation rewards $r_{1:T}$. Bottom: P-DAC, $i)$ encodes the (data-augmented) agent $o^a$ and expert $o^e$ trajectories into a latent space, $ii)$ passes them through the discriminator, and $iii)$ evaluates the DAC loss. $iv)$ Rewards $r_{1:T}$ can then be produced with the discriminator.

setting (Papagiannis & Li, 2020; Dadashi et al., 2021). A key component here is how we learn and compare latent representations using the cosine distance on a target encoder that is updated with the RL encoder's weights. Our second approach, *pixel discriminator actor critic*, P-DAC, is a GAIL-based method for pixels (Ho & Ermon, 2016; Kostrikov et al., 2019; Torabi et al., 2018). We propose several key modifications, including data augmentation and using the RL encoder for representations, that enable P-DAC to scale to non-trivial control tasks without needing access to expert actions. The imitation rewards from these methods are optimized with DrQ-v2 as an underlying image-based RL backbone (Yarats et al., 2021a).

We demonstrate the versatility and strong performance of both approaches on DeepMind control suite tasks, and find that both P-DAC and P-SIL are able to recover expert performance while outperforming canonical extensions of corresponding state-based IL methods, such as SIL (Papagiannis & Li, 2020) and DAC (Kostrikov et al., 2019; Ho & Ermon, 2016) in terms of performance and sample efficiency. Our extensive ablation study reveals insights into the key design decisions for pixel imitation learning. Finally, we also provide an easy-to-use implementation of these methods with unified DrQ-v2 backbone to make it simple for practitioners to build upon in this under-explored area.

## 2 BACKGROUND AND RELATED WORK

### 2.1 REINFORCEMENT LEARNING (RL) WITH PIXEL OBSERVATIONS

Pixel reinforcement learning can be instantiated as an infinite-horizon Markov decision process (MDP) (Bellman, 1957; Sutton & Barto, 2018), where the agent's state is approximated by a stack of consecutive RGB frames (Mnih et al., 2015). The MDP is of the form $(\mathcal{O}, \mathcal{A}, P, R, \gamma, d_0)$ where $\mathcal{O}$ is the observation space, $\mathcal{A}$ is the action space, $P : \mathcal{O} \times \mathcal{A} \to \Delta(\mathcal{O})$ is the transition function $(\Delta(\mathcal{O})$ is a probability distribution over $\mathcal{O})$, $R : \mathcal{O} \times \mathcal{A} \to \mathbb{R}$ is the reward function, $\gamma \in [0, 1)$ is the discount factor and $d_0$ is the initial state distribution. The RL problem consists of finding an optimal

policy $\pi : \mathcal{O} \rightarrow \Delta(\mathcal{A})$ that maximizes the expected long-term reward $\mathbb{E}_\pi[\sum_{t=0}^\infty \gamma^t R(\boldsymbol{o}_t, \boldsymbol{a}_t)]$, where $\boldsymbol{o}_0 \sim d_0, a_t \sim \pi(\cdot|\boldsymbol{o}_t)$ and $\boldsymbol{o}_{t+1} \sim P(\cdot|\boldsymbol{o}_t, \boldsymbol{a}_t)$.

## 2.2 IMITATION LEARNING

In imitation learning, agents do not have access to the environment reward $R$. Instead, they are provided with a dataset of multiple expert trajectories which the agent aims to imitate, where each trajectory is of the form $\boldsymbol{o}^e = (\boldsymbol{o}_0^e, \ldots, \boldsymbol{o}_T^e) \in \mathcal{O}^T$. More formally, agent trajectories $\boldsymbol{o}^a = (\boldsymbol{o}_0^a, \ldots, \boldsymbol{o}_T^a)$ under the policy $\pi$ have to be close to expert trajectories $\boldsymbol{o}^e$ under some metric between trajectories. We note these trajectories are empirical proxies for the occupancy distributions under the learner and expert policies (Ho & Ermon, 2016; Dadashi et al., 2021). This paper follows the line of research in imitation learning via agent-expert density matching, initially studied by Ng & Russell (2000); Boularias et al. (2011); Englert et al. (2013). Here, imitation-reward signals are derived from trajectories obtained by an expert policy; the agent is then trained using RL methods, which use these imitation rewards as learning signals.

**Generative Adversarial Learning for IL**  P-DAC builds on the budding area of state(-action)-based generative adversarial imitation learning, originally initiated by Ho & Ermon (2016) with the GAIL method, which uses adversarial learning combined with RL. A significant body of work built upon GAIL, notably AIRL and WGAIL (Fu et al., 2018; Xiao et al., 2019; Kostrikov et al., 2020), which propose alternative losses relying on other probabilistic divergences. Also, Torabi et al. (2018) extend GAIL to the no-action setting, a setting which we consider in this paper as well. DAC (Kostrikov et al., 2019) significantly improves upon GAIL by turning it into an offline algorithm, and adding a gradient penalty. Our P-DAC approach directly builds upon DAC, extending it to the visual setting. A key difference, which allowed us to stabilize visual DAC, was to store imitation rewards in the replay buffer and to not recompute them with up-to-date discriminators and encoders. Also, while DAC parameterizes the discriminator directly on the (proprioceptive) observation space, we compose the discriminator with the RL image encoder to leverage representations learned in the RL pipeline.

GAIL-based methods in visual spaces are nascent, and contain works leveraging auxiliary losses (Cetin & Celiktutan, 2021). Toyer et al. (2020) propose a robust visual IL benchmark in which they evaluate mainly adversarial approaches in the discrete action setting. Finally, Rafailov et al. (2021) concurrently propose a model-based extension of GAIL, which does adversarial learning in the latent space of a world model. As a baseline, they use a direct extension of DAC to visual spaces by parameterizing a discriminator in image space, which does not perform well on nearly all tasks they consider, while our P-DAC extension achieves expert performance on all considered tasks (including tasks considered by Rafailov et al. (2021), such as *walker walk* and *cheetah run*).

**Optimal Transport for IL**  P-SIL extends to the visual case recent SOTA works on imitation learning via optimal transport (OT), namely Dadashi et al. (2021) and Papagiannis & Li (2020). They both leverage optimal transport matching to define rewards. The former uses a greedy approximation of the Wasserstein while the latter leverages the entropic Wasserstein with cosine cost. However, the latter also learns an embedding for states adversarially, while we leverage the encoder of the RL algorithm, avoiding the need for minimax optimization, and allowing us to avoid any computational slowdown. Finally, most of the adversarial algorithms in the GAIL line (Ho & Ermon, 2016; Fu et al., 2018) can also be interpreted from the standpoint of the minimization of an OT functional between the state(-action) occupancy distribution of the agent and expert (Xiao et al., 2019). App. A reviews relevant background on optimal transport.

## 3 RECIPES FOR VISUAL IMITATION LEARNING

We now describe our IL approaches, which consist of alternative ways of defining imitation rewards that an RL backbone algorithm can learn from, and significantly improve upon canonical extensions of proprioceptive-state methods. In this paper, we leverage DrQ-v2 as the underlying RL algorithm (Yarats et al., 2021a). The data augmentation strategy throughout the paper (including for baselines) hence consists of random shifts with padding and a random crop to restore the original image dimension, followed by bilinear interpolation. For all methods, we gather episodes under the current policy, evaluate reward trajectories, and update the replay buffer with such episodes, replacing

environment rewards by imitation rewards. We summarize the overall process in algorithm 1. We also illustrate our two proposed methods in fig. 1.

## 3.1 Pixel Sinkhorn Imitation Learning (P-SIL)

Our first approach extends imitation learning algorithms based on optimal transport to a setting where only pixel observations are available. We define imitation-reward signals via the negative entropic Wasserstein distance between embedded agent and experts' image trajectories.

We interpret an image trajectory $\boldsymbol{o} = (\boldsymbol{o}_1, \ldots, \boldsymbol{o}_T)$ as a discrete probability measure of the form $\mu_{\boldsymbol{o}} = \frac{1}{T} \sum_{t=1}^{T} \delta_{\boldsymbol{o}_t}$, where images $\boldsymbol{o}_t \in \mathbb{R}^{C \times H \times W}$ are atoms weighted uniformly over time. Optimal transport distances directly on image observations provide weak signal given that $i$) the metric between individual images does not take into account spatial relationships between individual pixels, and $ii$) the sample complexity of OT grows exponentially with the number of dimensions (the number of samples in each trajectory necessary for a good estimate of the OT plan $\pi$) (Genevay et al., 2019).

To alleviate these challenges, we embed image trajectories using DrQ-v2's encoder:

$$\boldsymbol{o}^{a,\phi} = \left[ f_\phi(\boldsymbol{o}_1^a), \ldots, f_\phi(\boldsymbol{o}_T^a) \right], \qquad \boldsymbol{o}^{e,\phi} = \left[ f_\phi(\boldsymbol{o}_1^e), \ldots, f_\phi(\boldsymbol{o}_T^e) \right]. \tag{1}$$

In order to be agnostic to the scale of the encoded states, we consider the cosine distance as metric $d_c$ between encoded visual observations, similarly to Papagiannis & Li (2020),

$$C_{t,t'} = d_c(\boldsymbol{o}_t^{a,\phi}, \boldsymbol{o}_{t'}^{e,\phi}) = 1 - \frac{\langle \boldsymbol{o}_t^{a,\phi}, \boldsymbol{o}_{t'}^{e,\phi} \rangle}{\|\boldsymbol{o}_t^{a,\phi}\| \, \|\boldsymbol{o}_{t'}^{e,\phi}\|}. \tag{2}$$

We estimate the entropic Wasserstein distance with the cosine cost between embedded trajectories

$$\mathcal{W}_\epsilon^2(\mu_{\boldsymbol{o}^{a,\phi}}, \mu_{\boldsymbol{o}^{e,\phi}}) = \min_{\psi \in \Psi} \sum_{t,t'=1}^{T} C_{t,t'} \psi_{t,t'} - H(\psi), \qquad H(\psi) = - \sum_{t,t'=1}^{T} \psi_{t,t'} \log \psi_{t,t'}, \tag{3}$$

where $\Psi = \{\psi \in \mathbb{R}^{T \times T} : \psi \mathbf{1} = \psi^T \mathbf{1} = \frac{1}{T} \mathbf{1}\}$ is the set of coupling matrices, in order to obtain an optimal alignment $\psi^\star$. Finally, we extract rewards for each of the agent's states as

$$r(\boldsymbol{o}_t^{a,\phi}) = - \sum_{t'=1}^{T} C_{t,t'} \psi_{t,t'}^\star. \tag{4}$$

If we are provided with multiple expert trajectories $\boldsymbol{o}^{e_1}, \ldots, \boldsymbol{o}^{e_N}$, we infer the nearest-neighbor expert by computing the embedded entropic Wasserstein distance with cosine cost between each expert trajectory and the agent's rollout. We then set rewards with the alignment to the closest expert:

$$e_\star = \operatorname*{arg\,min}_{n \in \{1, \ldots, N\}} \mathcal{W}_\epsilon^2(\mu_{\boldsymbol{o}^{a,\phi}}, \mu_{\boldsymbol{o}^{e_n,\phi}}) \qquad r(\boldsymbol{o}_t^{a,\phi}) = - \sum_{t'=1}^{T} d_c(\boldsymbol{o}_t^{a,\phi}, \boldsymbol{o}_{t'}^{e_\star,\phi}) \psi_{t,t'}^\star. \tag{5}$$

The rewards computed in (5) are non-stationary as the encoder $f_\phi$ is updated to optimize the critic loss. To increase the stability of P-SIL, we use a target network $f_{\phi'}$, which is updated every $T_{\text{update}}$ environment steps with the weights of DrQ-v2's encoder. We use $f_{\phi'}$ to estimate rewards via (5). Also, we rescale P-SIL rewards so that the first episode is normalized to have a total return of $-10$ to account for rapid changes in encoder representations during initial training.

An important benefit of P-SIL is that computing imitation rewards does not add any computational burden as the cost of (5) is negligible, and we use the encoder of the RL agent. This is in contrast to adversarial approaches (Papagiannis & Li, 2020; Ho & Ermon, 2016), which require training a discriminator by solving an inner-loop maximization problem and then inferring rewards using the discriminator. We thus preserve the attractive computational efficiency of the base algorithm DrQ-v2.

In summary, we compute imitation rewards as the entropic Wasserstein discrepancy between the agent's and expert's encoded states for the single-demonstration case. For multiple expert demonstrations, we use the transport cost from the closest expert demonstration in the OT sense.

## 3.2 Pixel Discriminator Actor Critic (P-DAC)

We now propose and study P-DAC as a GAIL-based method for pixels that builds on the work by Ho & Ermon (2016); Torabi et al. (2018); Kostrikov et al. (2019). The key differences with previous works are that $i$) we compose the discriminator with the RL encoder to leverage its representations, $ii$) we apply data augmentation to discriminator inputs and $iii$) we store imitation rewards in the buffer rather than recomputing them at each iteration. We now describe the main components of P-DAC in more detail.

In contrast to P-SIL, it is necessary to train a discriminator, along with DrQ-v2's other networks (actor, critic and encoder). To train the discriminator, we sample batches $\boldsymbol{o}_{1:N}^e$ and $\boldsymbol{o}_{1:N}^a$ of size $N$ from both the expert buffer and the policy buffer, respectively, apply data augmentation, and encode augmented observations using DrQ-v2's encoder, which then yields:

$$\boldsymbol{o}^{a,\phi} = \Big[ f_\phi \circ \mathrm{aug}(\boldsymbol{o}_1^a), \ldots, f_\phi \circ \mathrm{aug}(\boldsymbol{o}_N^a) \Big], \qquad \boldsymbol{o}^{e,\phi} = \Big[ f_\phi \circ \mathrm{aug}(\boldsymbol{o}_1^e), \ldots, f_\phi \circ \mathrm{aug}(\boldsymbol{o}_N^e) \Big]. \quad (6)$$

Per common practice (Kostrikov et al., 2019), we omit importance sampling when estimating the policy state occupancy via replay buffer. We then maximize

$$\max_D \sum_{n=1}^N \log D(\boldsymbol{o}_n^{a,\phi}) + \sum_{n=1}^N \log(1 - D(\boldsymbol{o}_n^{e,\phi})) - \mathbb{E}_{\bar{\boldsymbol{o}}^\phi} \left[ \left\| \nabla D(\bar{\boldsymbol{o}}^\phi) \right\| - 1 \right]^2, \quad (7)$$

with respect to the discriminator's weights and add a gradient penalty as recommended by Kostrikov et al. (2019). Here, $\bar{\boldsymbol{o}}^\phi$s are sampled along straight lines between agent and expert's encoded observations. To compute rewards at the end of episodic rollouts, we embed the observation trajectory with DrQ-v2's encoder and then apply the discriminator to the encoded observations to obtain rewards

$$r(\boldsymbol{o}_t^a) = \log \Big( \sigma \circ D(\boldsymbol{o}_t^{a,\phi}) \Big) - \log \Big( 1 - \sigma \circ D(\boldsymbol{o}_t^{a,\phi}) \Big), \quad (8)$$

where $\sigma$ is the sigmoid function. We then add the episode with imitation rewards computed in (8) to the replay buffer. This differs from the original GAIL implementation (Ho & Ermon, 2016), where training is online, and from its follow-up offline extension DAC (Kostrikov et al., 2019) that recomputes rewards at each training iteration. We will show in the experiment section that data augmentation is a key component to scaling to challenging environments. Finally, in contrast with previous works that parameterize the discriminator directly in observation space (Ho & Ermon, 2016; Torabi et al., 2018; Kostrikov et al., 2019), we compose it with the RL encoder, which allows us to leverage its representations.

## 4 Experiments

In this section, we empirically evaluate the proposed algorithms on tasks from the DeepMind control suite, aiming to contrast their strengths and weaknesses. We also provide an extensive ablation study highlighting the key design choices that enable solving challenging tasks with imitation learning from visual observations only.

Our set of experiments was designed with the aim of answering the following questions:

1. *Are P-DAC and P-SIL able to achieve expert performance, and how do they compare in terms of sample efficiency (number of interactions with the environment required to solve a task) and computational efficiency?*
   Yes. P-DAC is more sample efficient than P-SIL, see fig. 2 while P-SIL has a 37% faster computational footprint than P-DAC, see fig. 3. Furthermore, our methods allow to learn policies that are closer to expert policies than baselines under the embedded Wasserstein metric, see fig. 4.

2. *Are P-DAC and P-SIL robust to the number of expert demonstrations they are provided?*
   Yes, see fig. 5.

3. *Is data augmentation as essential and effective as it is in pixel-based reinforcement learning?*
   Yes, agents cannot approach expert performance on nearly all task without data augmentation, see fig. 6.

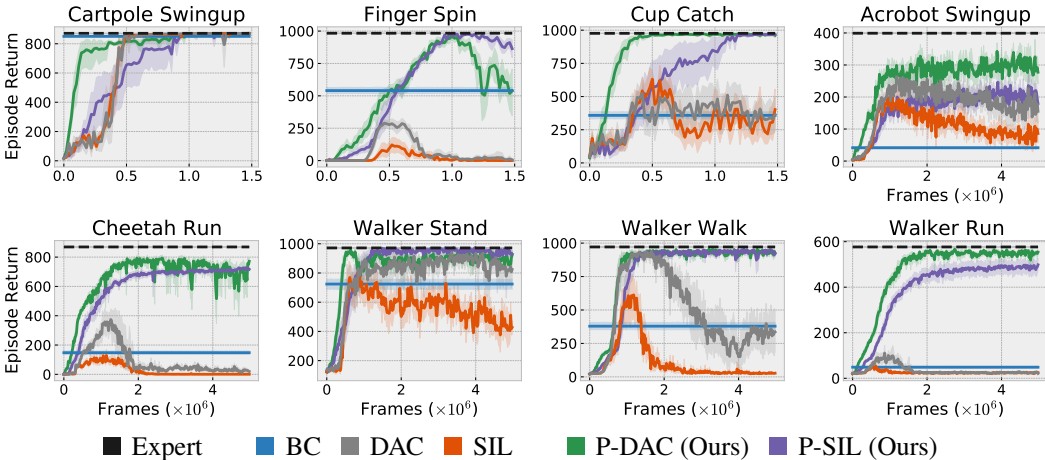

Figure 2: Our agents P-DAC and P-SIL demonstrate superior performance over vanilla instantiations of DAC and SIL, as well as BC over a set of challenging image-based tasks from DMC. In many cases P-DAC and P-SIL are able to recover the expert performance by using only 10 expert trajectories. Note, DAC and BC have privileged access to expert actions, while other methods do not.

4. *Is optimal transport providing a gain over simpler metrics without alignment (e.g., cosine distance between trajectories)?*

   Yes, there is a significant performance gain when leveraging OT alignments, see fig. 7

5. *Are expert actions required to solve challenging tasks or are pixel observations enough?*

   No, P-DAC without actions performs on par or better than P-DAC with actions on all environments, see fig. 8.

## 4.1 EXPERIMENTAL SETUP

**Environments**  We consider 8 Mujoco (Todorov et al., 2012) tasks in the DeepMind control suite (Tassa et al., 2018). The selected tasks are distinct enough to demonstrate the versatility and robustness of P-DAC and P-SIL. Environment observations are stacks of three consecutive $84 \times 84$ RGB images. We evaluate the agents with the environment rewards, but they are not provided to the agents during training. For the simplest tasks, we allow a budget of 1.5M environment frames, while for hard tasks we allow 5M environment frame in main results. In ablations, we allow 3M for hard tasks.

**Expert Demonstrations**  We gather expert demonstrations by training DrQ-v2 using the true environment rewards. We run 10 seeds and pick the seed that achieves highest episodic reward to generate expert trajectories (sequences of image observations).

**Baselines**  As baselines, we consider our two proposed approaches, P-DAC and P-SIL, along with a behavioral cloning (BC) baseline that we strengthen by leveraging the same data augmentation as in our approaches. We also consider canonical extensions of DAC (Kostrikov et al., 2019) and SIL (Papagiannis & Li, 2020), in which the discriminator is directly defined on pixel-space, and without the tricks proposed in this paper (e.g., no data-augmentation applied to discriminator inputs, no target encoder for SIL). DAC and SIL baselines are implemented with the same DrQ-v2 backbone as P-DAC and P-SIL, hence also benefit from data-augmentation in the RL training loop, thus making the comparison setup fair. We note that the BC and DAC baselines are privileged since they access expert actions while our baselines do not.

**Setup**  In all experiments, we run 10 seeds under each configuration and average results, while providing a 90% confidence interval. We compare agents with respect to two distinct scores. Firstly, we use the episodic return as a metric to verify whether the agent solves the task; secondly, we average the entropic Wasserstein distance with cosine cost between agent rollouts and expert demonstrations, both embedded via a common random encoder. This shows that agent rollouts are close to the expert

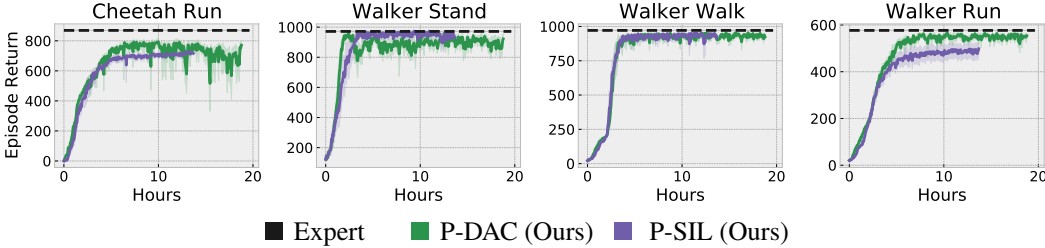

Figure 3: Performance of P-DAC, P-SIL, and data-augmented BC on DeepMind control suite tasks from pixel observations only with wall-clock time as axis. We notice P-SIL and P-DAC solve all tasks in less than 4 hours.

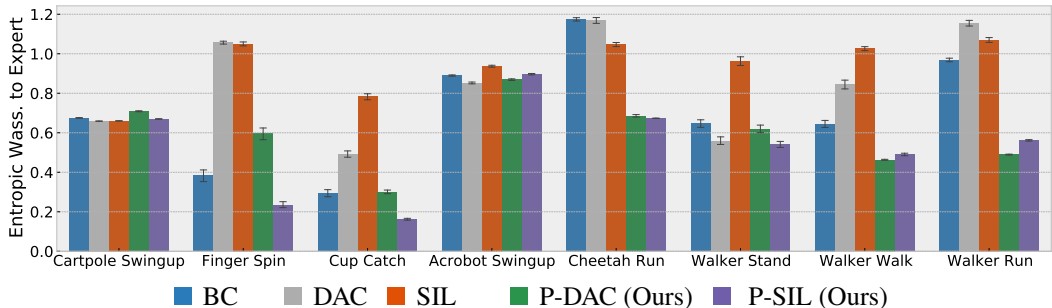

Figure 4: Entropic Wasserstein distance to the expert demonstrations (*the lower the better*) for P-DAC, P-SIL, SIL, DAC and data-augmented BC on DeepMind control suite tasks from pixel observations only, relative to a random policy. We notice P-SIL and P-DAC are on par in terms of closeness to expert demonstrations while BC, SIL and DAC are worse.

demonstrations they are trained on. Finally, in ablation bar-plots, we show episodic reward averaged over the final 10 environment episodes. App. D further describes all hyperparameters.

## 4.2 MAIN RESULTS

**Sample efficiency** We begin by comparing baselines with respect to sample efficiency, i.e., the number of environment interactions required to solve a task. We observe in fig. 2 that both P-DAC and P-SIL achieve expert performance on all environments besides acrobot swingup, which is a hard exploration task. Both approaches significantly surpass the performance of SIL, DAC and data-augmented behavioral cloning even though the latter two are privileged by knowing expert actions. We also note that P-DAC is more sample efficient than P-SIL.

In fig. 4, we show the entropic Wasserstein between embeddings of agent rollouts and expert trajectories; results correlate well with those under episodic return (see fig. 2). Again, P-SIL and P-DAC outperform data-augmented BC under this alternative metric, i.e., trajectories under these are closer in the embedded entropic Wasserstein sense than those of BC.

**Computational efficiency** We contrast the computational efficiency of our approaches in fig. 3; the comparison is fair as methods are implemented on a unified backbone. Both of our approaches solve the hard tasks in around 4 wall-clock hours on a single V100 GPU. This can be explained by the fact that there is no loss in frames per second (FPS) for P-SIL over its RL backbone ($\text{FPS}_{\text{PSIL}} = 101$), while P-DAC's FPS is impacted by the cost coming from discriminator training ($\text{FPS}_{\text{PDAC}} = 74$). As a result, both approaches have comparable computational efficiency.

## 4.3 ABLATIONS

We now provide an extensive ablation contrasting the approaches and highlighting key design choices.

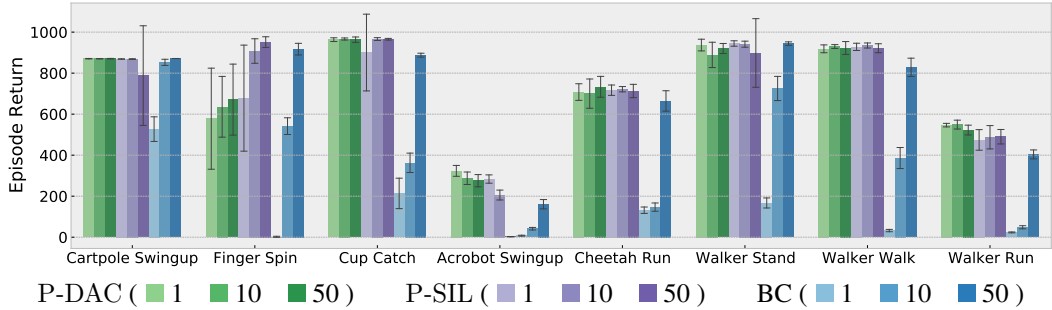

Figure 5: Ablation study on the number of demonstration. We train all P-DAC, P-SIL and BC on $1, 10$ and $50$ demonstrations. We observe P-DAC and P-SIL are robust to the number of demonstrations, while BC is not, and fails on most tasks for the 1-demonstration setting.

**Number of demonstrations** We first evaluate the robustness of baselines to the number of expert demonstrations provided to the learner. We train agents on $1, 10$ and $50$ demonstrations, on all tasks, and plot episodic return averaged across seeds in fig. 5. We observe that our approaches are robust to the number of demonstrations, with comparable performance for $1, 10$ and $50$ demonstrations across tasks (besides finger spin). By contrast, the data-augmented BC baseline is non-robust, and requires a large number of demonstrations (typically $50$) to approach expert performance.

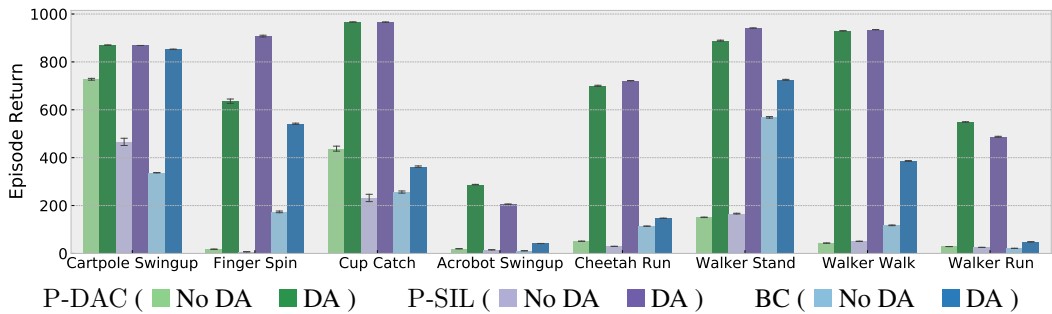

Figure 6: Ablation study on data augmentation (DA). We compare P-DAC, P-SIL and BC with and without data augmentation. We observe the gap in performance between data-augmented (DA) and no-data-augmented (No DA) implementations to be large. For instance, our baselines do not recover any meaningful behaviors on finger spin, cheetah run, acrobot swingup, and walker environments without DA.

**Data augmentation** Next, we evaluate whether data augmentation is necessary to achieve expert performance. We observe that without augmentation, both P-DAC and P-SIL are not able to recover solid performance on nearly all tasks, especially locomotion ones. We note that behavioral cloning is less sensible to it. This may be explained by the fact that augmentation is necessary to obtain good representations as seen in Laskin et al. (2020); Yarats et al. (2019; 2021a), and that our imitation approaches leverage such representations to design reward signals. As a result, without good representations the agents can not recover any meaningful behaviors.

**Metrics** We continue with analyzing one of the key design choices behind P-SIL, i.e., whether an optimal transport alignment of the agent and expert trajectories is required over simply leveraging an $l_2$ or cosine distance (no alignment). We observe in fig. 7 that OT is essential to achieving expert performance on most tasks. We believe this is due to the high variance in the initial state distribution of agent and expert, which leads to unaligned trajectories, and hence uninformative rewards in the absence of OT alignment.

**Observations vs. Observation-actions** GAIL-like methods are often assumed to require expert actions to recover non-trivial performance. We hence compare P-DAC with a privileged implementa-

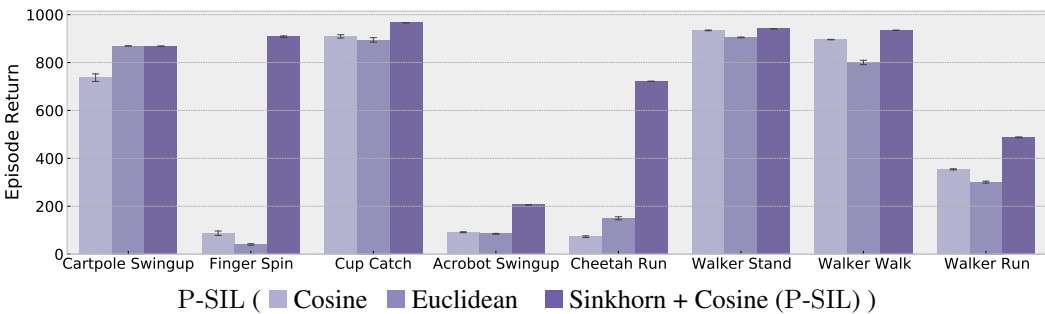

Figure 7: Ablation study on the distance between trajectories for P-SIL. We compare our approach (OT alignment with cosine cost), to the cosine distance and the Euclidean distance between trajectories (without OT). We observe that OT significantly outperform non-OT approaches. In particular, our baselines without OT do not recover any meaningful behaviors on finger spin and cheetah run, and are weaker on acrobot swingup and walker run. Hence, OT alignment is also an essential component of P-SIL.

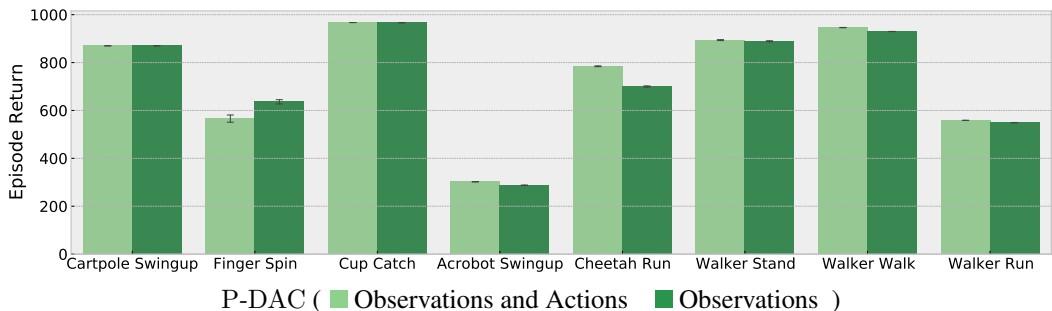

Figure 8: Ablation study on the need for expert actions. We train P-DAC with and without actions concatenated to observations (the former is privileged). We observe the performance gap to be marginal, hence actions are not necessary in the considered environments, contrarily to popular beliefs on GAIL-like algorithms.

tion of P-DAC which accesses expert actions. In the latter case, we concatenate embedded image observations, and actions before passing these to the discriminator. We observe in fig. 8 that our approach without expert actions performs on par with the action-baseline, highlighting that on the considered environments, actions are not required to achieve strong performance. This is encouraging given that there are massive amounts of unlabeled video data that these agents could potentially learn from.

## 5 CONCLUSION

We propose effective methods for imitation learning directly from pixels without expert actions, extending state-based adversarial and optimal transport approaches, and significantly outperforming canonical extensions of these. This steps us closer toward leveraging the rich amount of expert-level data on non-trivial environments for the control of continuous systems. We envision that continuing to scale in this space will involve further understanding how to recover a representation of the high-dimensional data, *e.g.* with unsupervised pretraining, and how to go beyond settings where the expert and learner live in the same space.

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

## A  BACKGROUND ON OPTIMAL TRANSPORT

Optimal transport (Villani, 2009; Peyré & Cuturi, 2019) tools allow us to compare probability measures while incorporating the geometry of the space. The entropic 2-Wasserstein distance between two discrete measures $\mu_{\boldsymbol{x}} = \frac{1}{T} \sum_{t=1}^{T} \delta_{\boldsymbol{x}_t}$ and $\mu_{\boldsymbol{y}} = \frac{1}{T} \sum_{t=1}^{T} \delta_{\boldsymbol{y}_t}$ is

$$\mathcal{W}_\epsilon^2(\mu_{\boldsymbol{x}}, \mu_{\boldsymbol{y}}) = \min_{\psi \in \Psi} \sum_{t,t'=1}^{T} d^2(\boldsymbol{x}_t, \boldsymbol{y}_{t'}) \gamma_{t,t'} - H(\pi), \qquad H(\psi) = -\sum_{t,t'=1}^{T} \psi_{t,t'} \log \psi_{t,t'}, \qquad (9)$$

where $\Psi = \{\psi \in \mathbb{R}^{T \times T} : \psi \mathbf{1} = \psi^T \mathbf{1} = \frac{1}{T} \mathbf{1}\}$ is the set of coupling matrices, and $d$ is a metric. Adding the entropic term has two main benefits: $i)$ the induced problem can be solved in quadratic time via Sinkhorn's algorithm (Sinkhorn & Knopp, 1967), and $ii)$ the resulting distance is smooth in the measures' samples. Intuitively, the optimal coupling matrix $\psi$ provides an alignment of the samples of $\mu_x, \mu_y$ which minimizes the cost of transport between these. The distance then consists of the weighted sum of distances between aligned samples, along with an entropic penalty.

## B  DRQ-V2

We use DrQv2 (Yarats et al., 2021a), which is a actor-critic method for continuous control based on the deep deterministic policy gradient (DDPG) (Lillicrap et al., 2015). Given a replay buffer $\mathcal{D}$, it learns simultaneously a $Q$-function $Q_\theta$ and a policy $\pi_\eta$. $Q_\theta$ is trained by clipped double-$Q$-learning (Fujimoto et al., 2018) with $n$-step returns. $\pi_\eta$ is trained via deterministic policy gradient. DrQv2 employs data augmentation in the form of random shifts with padding and a random crop to restore the original image dimension, followed by bilinear interpolation. Data augmentation acts as regularization and reduces the variance of the $Q$ estimates. Images are embedded into the latent space via an encoder $f_\phi$ after being augmented. This encoder is trained to minimize the critic loss only.

## C  ALGORITHMS

---

**Algorithm 1** Inverse reinforcement learning core. Different methods can be instantiated by changing the rewarder function.

---

**Require:** Expert demonstrations $\{\mu_{\boldsymbol{o}^{e_n}}\}_{n=1}^{N}$, replay buffer $\mathcal{D}$, initialized policy network $\pi$, Q-network $Q$ and encoder $f$. For DAC-like baselines, also requires a discriminator $D$.

   **for** $t \in T_{total}$ **do**
      **if** done **then**
         $r_{1:T} = \text{rewarder}(\text{episode})$, for instance (5) for P-SIL, (8) for P-DAC
         Update episode with $r_{1:T}$ and add all quadruples $[\boldsymbol{o}_t, \boldsymbol{a}_t, \boldsymbol{o}_{t+1}, r_t]$ to $\mathcal{D}$.
         $\boldsymbol{o}_t = \text{env.reset()}$, done = False, episode = $[\,]$
      **end if**
      $\boldsymbol{a}_t \sim \pi(\cdot|\boldsymbol{o}_t) \to \boldsymbol{o}_{t+1}$, done = env.step$(\boldsymbol{a}_t)$, episode.append$([\boldsymbol{o}_t, \boldsymbol{a}_t, \boldsymbol{o}_{t+1}])$
      Update DrQ-v2's actor and critic, and rewarder-specific functions using $\mathcal{D}$.
   **end for**

---

## D   HYPERPARAMETERS AND FURTHER EXPERIMENTAL DETAILS.

We provide a list of all hyperparameters in table 1. The only environment-specific variation is that the number of $n$-steps is set to 1 for walker tasks. For P-DAC, we use the DrQ-v2's critic encoder, while for P-SIL we use the actor's encoder, noting that both share the same convolutional layer weights.

| Agent | Parameter | Value 1 |
|---|---|---|
| Common | Replay buffer size | 150000 |
| | Learning rate | $1e^{-4}$ |
| | Exploration Schedule | $\mathrm{linear}(1, 0.1, 500000)$ |
| | Discount | 0.99 |
| | $n$-step returns | 3 |
| | Action repeat | 2 |
| | Frame stack | 3 |
| | Seed frames | 2000 |
| | Exploration steps | 4000 |
| | Mini-batch size | 256 |
| | Agent update frequency | 2 |
| | Critic soft-update rate | 0.01 |
| | Features dim | 50 |
| | Hidden dim | 1024 |
| | Optimizer | Adam |
| P-SIL | Target update frequency | 10000 |
| | Reward scale factor | 10 |
| P-DAC | Gradient penalty coefficient $\lambda$ | 10 |

Table 1: List of hyperparameters.

# E EXTRA ABLATION PLOTS

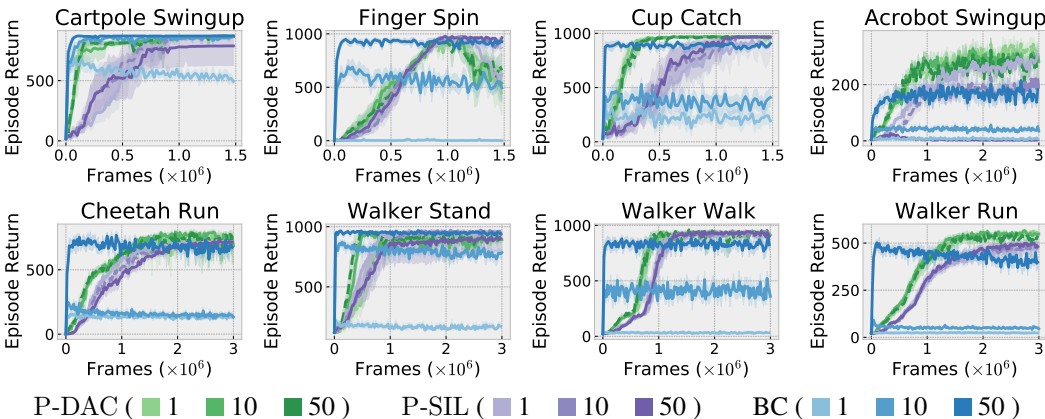

Figure 9: Ablation study on the number of demonstration. We train all P-DAC, P-SIL and BC on 1, 10 and 50 demonstrations. We observe P-DAC and P-SIL are robust to the number of demonstrations, while BC is not, and fails on most tasks for the 1-demonstration setting.

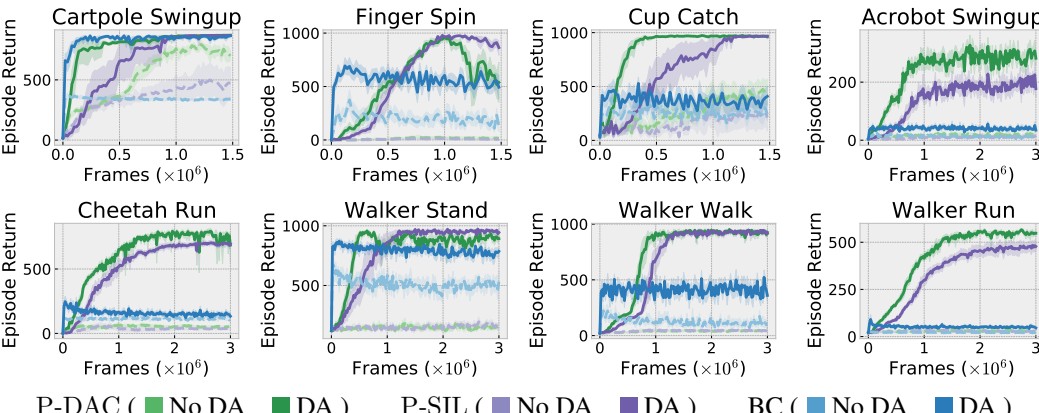

Figure 10: Ablation study on data augmentation (DA). We compare P-DAC, P-SIL and BC with and without data augmentation. We observe the gap in performance to be significantly larger for P-DAC and P-SIL, showing that DA is a crucial component of our methods.

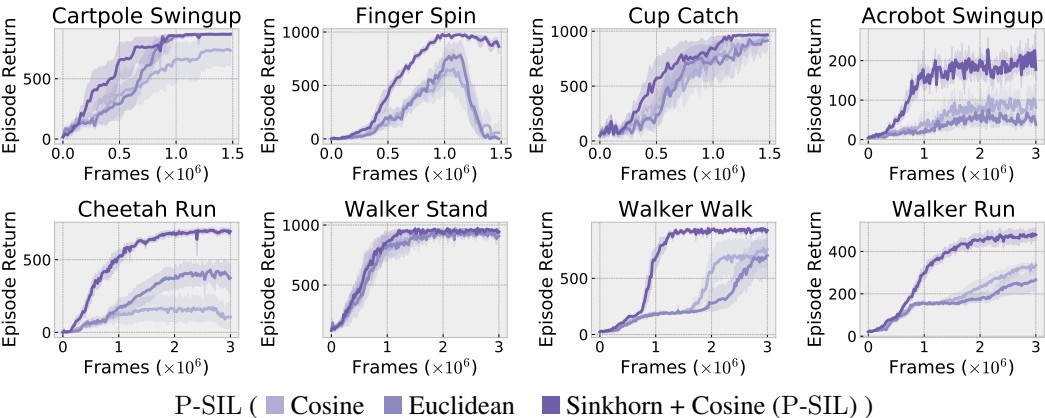

Figure 11: Ablation study on the distance between trajectories for P-SIL. We compare our approach (OT alignment with cosine cost), to the cosine distance and the Euclidean distance between trajectories (without OT). We observe that OT significantly outperform non-OT approaches. Hence, OT alignment is also an essential component of P-SIL.

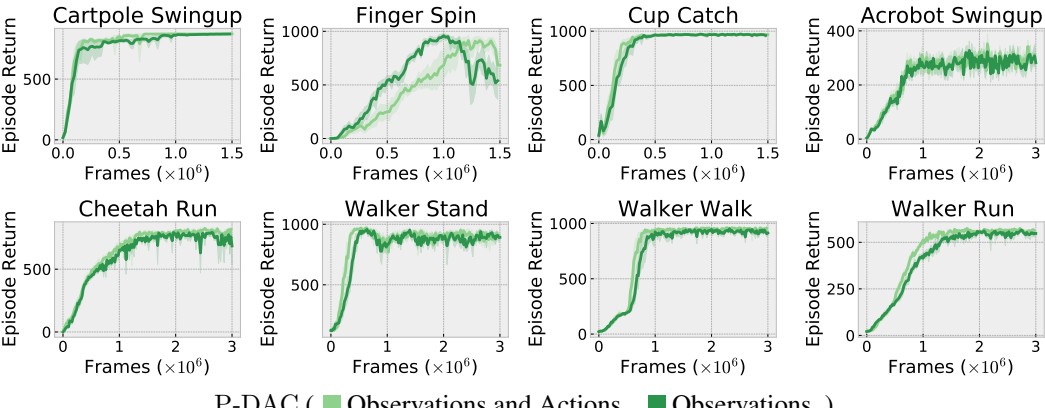

Figure 12: Ablation study on the need for expert actions. We train P-DAC with and without actions concatenated to observations (the former is privileged). We observe the performance gap to be marginal, hence actions are not necessary in the considered environments, contrarily to popular beliefs on GAIL-like algorithms.

