# OpenReview forum: "Imitation Learning from Pixel Observations for Continuous Control"
_ICLR.cc/2022/Conference — ICLR 2022 Submitted_

### Official Review · Reviewer_JN38 · 2021-10-28

**Correctness:** 3
**Technical Novelty And Significance:** 2
**Empirical Novelty And Significance:** 2
**Recommendation:** 5
**Confidence:** 4

**Main Review:**

Strength:
+ The paper is well-motivated. It is attractive to enable agents to learn from demo videos without actions.
+ The paper is well-written and easy to follow.
+ Two approaches (P-SIL and P-DAC) are proposed based on stinkhorn imitation learning (SIL) and discriminator actor-critic (DAC), respectively. The introduced data augmentation method is simple yet effective in improving performance.
+ The designed experiments comprehensively answered common questions on effectiveness, efficiency, robustness, and necessity.

Weakness:
- The setting is impractical. The proposed methods rely on an RL expert for computing the reward. However, such an RL expert is unavailable in most video demos. Notably, if the RL expert is available, it will be convenient to access the expert action for training. This is my main concern for this paper.
- It seems that the data augmentation method contributes most to the improvement in the performance. How about the performance of DAC and SIL with the data augmentation methods?


**Summary Of The Paper:**

This paper focuses on learning visual policies by imitating video data without observing the expert actions. The authors introduce a method to compute the imitation reward based on the representation from the expert RL encoder. Besides, a data augmentation method is proposed to scale to non-trivial control tasks. They conduct comprehensive experiments on DeepMind control suite tasks, showing the versatility and strong performance of the two approaches (P-DAC and P-SIL).

**Summary Of The Review:**

My main concern for this paper is the setting. In my view, it is unreasonable to use an encoder from the RL-based expert for learning from videos without actions.

---

> ### Author Response · Authors · 2021-11-17
> **Rebuttal**
>
> Thank you for your time reviewing our paper and giving feedback! We are glad you considered our work well-motivated, well-written and easy to follow, along with an attractive setting. We would like to clarify some points of misunderstanding on some of the experimental details, and address the concerns raised.
>
> **It seems that the data augmentation method contributes most to the improvement in the performance. How about the performance of DAC and SIL with the data augmentation methods?:**
> We emphasize that all imitation baselines (besides BC), including DAC and SIL, P-DAC and P-SIL have a unified DrQ-v2 backbone, hence all leverage the same method for data augmentation (i.e., padding, random crops, and bilinear interpolation). We further clarify this point at the bottom of page 3, and in the baselines paragraph on page 6. As a result, the empirical comparison to DAC and SIL is fair, which highlights that an important contributor to the performance is instead the way we perform representation learning, notably that we leverage representations from the RL encoder to embed visual observations prior to performing OT  (for P-SIL) or applying the discriminator (for P-DAC), along with various other proposed tricks. The requested experiment is hence Figure 2 in the paper.
>
> **The setting is impractical. The proposed methods rely on an RL expert for computing the reward. However, such an RL expert is unavailable in most video demos:**
> We agree the setting that you are describing is highly impractical, that is why we **do not** use it. We would like to emphasize that our setup does not require an RL expert. We only require access to a fixed number (between 1 and 50) of pixel image trajectory from an expert (o_1,...,o_T) where o_t is an image for all t. This is the canonical setup of imitation learning from observation in the field (see GAIL, DAC), which we adhere to. The agent is trained using these expert demonstrations only, via reinforcement learning. In particular, we learn an encoder online in the training loop. Furthermore, we make this setup even more realistic by dropping the access to expert actions and only requiring observations.
>
> As a summary, the comparison setting is fair as baselines are implemented with the same SOTA backbone DrQ-v2 and the same data augmentation strategy. Also, this setting is widely used in imitation learning, and is the basis for methods like GAIL [1], DAC [2].
>
> [1] Jonathan Ho and S. Ermon. Generative adversarial imitation learning. In NeurIPS, 2016.
>
>
> [2] Ilya Kostrikov, Kumar Krishna Agrawal, Debidatta Dwibedi, Sergey Levine, and Jonathan Tompson. Discriminator-actor-critic: Addressing sample inefficiency and reward bias in adversarial imitation learning. In International Conference on Learning Representations, 2019.

---

> > ### Comment · Reviewer_JN38 · 2021-11-28
> > **Response to Rebuttal**
> >
> > I thank the authors for providing detailed answers to my questions. Even though the rebuttal addressed some misunderstandings, but I am still concerned about the marginal contribution of this paper, after reading the response from other reviewers. So I prefer to keep the original score.

---

### Official Review · Reviewer_wukE · 2021-10-29

**Correctness:** 4
**Technical Novelty And Significance:** 2
**Empirical Novelty And Significance:** 2
**Recommendation:** 5
**Confidence:** 4

**Main Review:**

Pros:

- Easy to implement and understand algorithms, albeit knowing the algorithms they are built upon.

- Good results on pixel-based imitation learning

- Open-source implementation is provided.

##########################################################################

Cons:

- Adding data augmentation and DrQ-v2 as backbone is a sound and effective approach but there is little novelty in this paper, as auxiliary losses and target encoders to improve pixel observations have been studied before. To my understanding the contributions for P-SIL with respect to SIL are using the DrQ-v2 RL encoder to embed states instead of an adversarial approach. Instead for P-DAC, the contributions with respect to DAC are again composing it with the DrQ-v2 RL encoder and computing the rewards at the end of episodic rollouts instead of recomputing them at each training iteration.

- Paper is hard to read:
    - pseudocode of algorithm is not detailed enough to understand your contribution. You could expand on the `rewarder` function which should be your actual contribution.
    - pseudocode of algorithm doesn't have any initialization and definition of variables.
    - in equation 3, $\Psi$ and $\psi$ are not defined.
    - a background on DrQ-v2 would be highly appreciated to understand how the training pipeline works and the gradient flows, which I had to hypotesize after reading DrQ and DrQ-v2.

##########################################################################

Questions during rebuttal period:

- Address and/or clarify the cons above.

**Summary Of The Paper:**

This paper presents two algorithms for imitation learning P-SIL and P-DAC which are built on top of SIL (Papagiannis & Li, 2020) and DAC (Kostrikov et al, 2019). The first one is an algorithm based on the Sinkhorn distance and the second one uses an adversarial approach, by training a discriminator. These two approaches use state-only trajectories to generate reward sequences to match the agent state-trajectories and learn through that reward. By combining the advances in image-based reinforcement learning by the usage of data augmentation and target encoders, the two new algorithms, P-SIL and P-DAC, are tested and reach good results on visual control tasks.


**Summary Of The Review:**

I am towards rejection of this paper. Sound and interesting approach which achieves good results, but there is little novelty. Furthermore, paper is quite hard to read and to understand what are the main contributions, which makes an approach that seems easy quite complex.

---

> ### Author Response · Authors · 2021-11-17
> **Rebuttal**
>
> Thank you for your time reviewing our paper and giving feedback! We are glad you agreed that we achieve good results on pixel-based IL, and that we propose easy-to-implement algorithms. We incorporated the clarifications you required in the paper (in red), and will address the comment on the novelty of our approaches.
>
> **in equation 3, Ψ and ψ are not defined:**
> Thank you for catching this. We have updated the paper to define these below eq (3). $\Psi$ is the set of couplings (doubly-stochastic matrices), marginalizing to uniform discrete measures. This is also described in Appendix A.
>
> **a background on DrQ-v2 would be highly appreciated to understand how the training pipeline works and the gradient flows, which I had to hypotesize after reading DrQ and DrQ-v2:**
> We agree, and have added background on DrQ-v2 in Appendix B.
>
> **Adding data augmentation and DrQ-v2 as backbone is a sound and effective approach but there is little novelty in this paper, as auxiliary losses and target encoders to improve pixel observations have been studied before:**
> While auxiliary losses, target encoders and data augmentation have been studied in the pixel-based reinforcement learning setting, they have not been explored much in the imitation setting, hence making it a contribution to our paper. Concurrent work [1] accepted at NeurIPS 2021 implements a data-augmented version of DAC for pixel-based IL as baseline (with access to expert actions), and it is not solving even simple tasks like walker-walk, which our approaches P-SIL and P-DAC manage to solve successfully. We recover the same empirical observation as [1] with our data-augmented DAC baseline, which collapses for walker-walk too. This illustrates one of our main contributions which consists in figuring out that leveraging representations from the RL encoder to define rewards stabilizes training and allows to achieve expert performance in challenging environments.
>
> **Pseudocode of algorithm:**
> We added initialization and definition of variables in the updated version of the paper (in red). We also added a reference for the rewarder to the reward functions used for our baselines P-SIL and P-DAC.
>
> [1] Rafael Rafailov, Tianhe Yu, Aravind Rajeswaran, and Chelsea Finn.  Visual adversarial imitation learning using variational models. In NeurIPS, 2021.

---

> > ### Comment · Reviewer_wukE · 2021-11-19
> > **Question on contribution**
> >
> > Thanks for your reply. The updated manuscript clarifies some of the concerns I had. Although I am still unsure of the actual contribution of this work. Is it the usage of SIL and DAC with pixel based observations and the addition of data augmentation? From my understanding, SIL and DAC were already doing imitation learning from observations only and your addition is basically joining SIL and DAC with DrQ-v2.

---

> > > ### Author Response · Authors · 2021-11-24
> > > **Reply on contributions**
> > >
> > > Thanks a lot for your reply. To further clarify, SIL and DAC baselines in the paper already leverage data augmentation (which we added to make the setup fair to our proposed methods P-SIL and P-DAC).
> > >
> > > The main difference between P-SIL and SIL is the representations we feed to the OT computation. SIL proposed learning a discriminator adversarially, and to feed representations from the discriminator to the Sinkhorn loss. By contrast, we leverage the representations learned in the RL loop, hence we do not need to learn any extra network with an extra optimisation problem. In the SIL paper, the approach is applied to the case of proprioceptive state-based learning. We show in our paper that naively extending SIL to pixel-based observations is not effective (see Fig 2), by contrast with our approach P-SIL.
> > >
> > > The main difference between P-DAC and DAC is similar. In DAC, the discriminator is defined directly on pixel observations, while in P-DAC we compose a small discriminator head with the encoder trained in the RL loop, allowing to leverage representations learned by RL, and simplifying the adversarial optimisation problem. While we leverage data augmentation in the RL loop for both DAC and P-DAC, we also motivate leveraging data augmentation in the discriminator training, which also brings a performance boost.
> > >
> > > We also propose various other modifications including appending rewards to the replay buffer in an offline manner, which we’ve observed to stabilise training.
> > >
> > > **your addition is basically joining SIL and DAC with DrQ-v2**
> > >
> > > Also about this comment, the baselines (DAC and SIL) we compare our approaches (P-DAC and P-SIL) to also use DrQ-v2 as backbone, to make the setup fair, hence we believe the main contributions lie in the above points rather.
> > >
> > > We hope this clarifies your concern, and would be happy to further discuss the main contributions of the paper.

---

> > > > ### Comment · Reviewer_wukE · 2021-11-29
> > > > **Thanks for the reply**
> > > >
> > > > I appreciate the detailed answer. However, I do not feel that the contribution is clear on the paper. For this reason I am not changing my score.

---

### Official Review · Reviewer_omV5 · 2021-11-02

**Correctness:** 2
**Technical Novelty And Significance:** 3
**Empirical Novelty And Significance:** 2
**Recommendation:** 3
**Confidence:** 4

**Main Review:**

STRENGTHS

(S1) The stated thrust of the paper--that of exploring the design choices and tradeoffs within the space of algorithms designed for imitation learning from visual observations--is of great importance to the community studying these methods. To date, I can think of no paper that provides a compelling discussion of these topics.

(S2) The authors have provided an open-source implementation of their methods, which bolsters the story of the paper being one of exploring "recipes" for visual imitation.

WEAKNESSES

(W1) Unfortunately, I don't feel that the authors really lived up to the goals that they stated at the outset of the paper. That is, even with proposing and experimenting with P-SIL and P-DAC, I still don't have a clear understanding of the "design choices and tradeoffs" in this algorithmic space. Nowhere in the paper can I find a clear and explicit discussion of these things--the five experimental questions stated at the beginning of Section 4 seem to ultimately focus more on whether or not P-SIL and P-DAC will perform well, as opposed to quantifying or discovering which design choices are more important. For example, with respect to data augmentation, Figure 6 really only serves to show that P-DAC and P-SIL are better than BC rather than exploring the effect of data augmentation as a specific design choice across the suite of available methods. Moreover, with respect to the experiment depicted in Figure 7, it does not seem to me that the presented data matches up with the conclusion. Many of the domains show that all the methods compared perform about as well no matter what distance function is used--this doesn't at all support the claim that the authors have made that "OT alignment significantly outperform [sic] non-OT approaches." Finally, if P-SIL and P-DAC are both built atop DrQ-v2 and the other methods compared against are not, what evidence is there that the performance gain is not simply due to the use of DrQ-v2 as opposed to the "representation sharing" and other more general factors that the authors claim account for this difference? I do note that there _is_ unused space at the end of Page 9--I wonder if the authors might be able to address this comment with an enhanced Conclusion for example.

(W2) Again, for a paper purporting to discuss design choices and tradeoffs, I was disappointed that more experimental details weren't given. For example, with respect to the data augmentation experiments, I could not find anywhere in the paper where it was specified what exact types of data augmentation were performed for these experiments.

MINOR COMMENTS

(MC1) The bottom half of Figure 1 failed to render in an application on my iPad, though it did render correctly in Chrome. Perhaps the authors could pursue a more reliable way to generate that figure to avoid this problem in the next version of the paper.

**Summary Of The Paper:**

The authors have considered the problem of imitation learning using only visual observations, and they've explored the paradigms of using methods based on adversarial learning and optimal transport to solve this problem. In particular, the authors propose two new methods--P-SIL and P-DAC--presumably as a way to explore design choices and tradeoffs in this space. Some experiments designed to demonstrate the superiority of P-SIL and P-DAC are presented.

**Summary Of The Review:**

I agree wholeheartedly with the goal that the authors have stated in the early portions of the paper, but I think the work that has been done--or at the very least, the way it has been presented--fails to live up to that goal. I'd encourage the authors to focus more on the design decisions and tradeoffs in the existing space of algorithms rather than trying to show that their proposed algorithms dominate others.

---

> ### Author Response · Authors · 2021-11-17
> **Rebuttal**
>
> Thank you for your time reviewing our paper and giving feedback! We are delighted you agreed the problem tackled is of great importance to the community studying these methods, and that no compelling discussion/study of this setting has already been performed. We will now aim to address points of concerns/misunderstandings, in particular on the fairness of the experimental setup, and on the significance of the results.
>
> **Figure 6 really only serves to show that P-DAC and P-SIL are better than BC rather than exploring the effect of data augmentation as a specific design choice across the suite of available methods:**
> Rather than showing that P-DAC and P-SIL are better than BC, the goal behind Figure 6 was to show that data augmentation is an essential component of our methods, i.e., that when turning it off (No DA in the plot), the performance significantly decreased. We indeed observe in this figure that in finger-spin, cheetah-run, acrobot-swingup and walker environments, P-DAC and P-SIL without DA do not recover any meaningful behavior. We clarified this in the figure’s caption. Finally, the reason we decided to ablate DA for BC too is so the study is fair.
>
> **Figure 7, it does not seem to me that the presented data matches up with the conclusion:**
> In Figure 7, we can observe that baselines without OT alignment completely fail on finger-spin and cheetah-run while with our baseline with OT alignment solves both tasks. Also, on acrobot-swingup and walker-run, the performance gap ranges between 30% and 50% when comparing non-OT and OT baselines. For these reasons, we had concluded in the text that OT significantly outperformed non-OT approaches. We clarified this in the figure’s caption, notably detailing on what environments we noticed significant gains.
>
> **Finally, if P-SIL and P-DAC are both built atop DrQ-v2 and the other methods compared against are not, what evidence is there that the performance gain is not simply due to the use of DrQ-v2 as opposed to the "representation sharing" and other more general factors that the authors claim account for this difference?:**
> We emphasize that all imitation baselines (besides BC), including DAC and SIL, P-DAC and P-SIL have a unified DrQ-v2 backbone, hence all leverage the same method for data augmentation (i.e., padding, random crops, and bilinear interpolation). As a result, the empirical comparison to DAC and SIL is fair. We further clarify this point at the bottom of page 3, and in the baselines paragraph on page 6.
>
> **For example, with respect to the data augmentation experiments, I could not find anywhere in the paper where it was specified what exact types of data augmentation were performed for these experiments:**
> We will clarify in the paper that we leverage the DrQ-v2 data augmentation, which consists of random shifts with padding and a random crop to restore the original image dimension, followed by bilinear interpolation. We clarified this at the bottom of page 3.

---

> > ### Comment · Reviewer_omV5 · 2021-11-19
> > **Questions Remain**
> >
> > Thanks to the authors for taking the time to respond to the points above. While I thank the authors for providing extra detail regarding the data augmentation method used, I unfortunately do not feel as though they have adequately addressed my other major concerns.
> >
> > With respect to Figure 6, I'm afraid I still don't believe that the experiment answers the question posed at the beginning of Section 4, i.e., "is data augmentation as essential and effective as it is in pixel-based reinforcement learning?" To adequately answer this question in the context of the present paper, it seems to me that the authors ought to show how _all_ the methods do with and without data augmentation. For example, how does DAC perform once data augmentation is added? The answer to this question is important to be able to answer the stated question.
> >
> > With respect to Figure 7, I think the authors are on the right track to acknowledge that OT really only seems to outperform the cosine and Euclidean baselines on certain domains, but the real question here is: why? The authors make a very strong claim in the paper that OT is "essential to achieving expert performance on most tasks," but I don't think the claim is true as stated. Moreover, without some reasonable intuitive explanation for why OT is essential for some domains but makes no difference in others, I remain skeptical of the claim that OT is somehow a general and critical component in a recipe for IL from pixel observations.
> >
> > With respect to the point about DrQ-v2, I'm afraid I'm not sure how the author response or the changes in the paper have addressed my comment. To reiterate, the paper makes the claim that there are general algorithmic components that are essential for successful IL from pixel observations. These are: using encoded representations in an AIL framework, performing data augmentation, and using an optimal transport distance metric. If this claim is true, then it should not depend on the use of DrQ-v2 specifically. Rather, it should apply using _other_ RL backbones as well. However, the authors have not provided any experiment or discussion to convince the reader that this is true.

---

### Official Review · Reviewer_R6AN · 2021-11-03

**Correctness:** 3
**Technical Novelty And Significance:** 2
**Empirical Novelty And Significance:** 2
**Recommendation:** 5
**Confidence:** 3

**Main Review:**

Pros:

Two imitation learning methods based on RL encoder are proposed. The experimental results show that both of the methods outperform the DAC and SIL baselines.

Cons:
1. Data augment is important for improving the performance of P-DAC. However, it is not clear how to augment the data in practice.
2. It is not always possible to used the RL encoder for many tasks. This is the key issue of this work.
3. The It is necessary to compare the proposed methods with other extentions of DAC and SIL.

**Summary Of The Paper:**

This paper introduces two methods for imitation learning from pixel observations based on adversarial learning and optimal transport. Representation from the RL encoder are adopted for calculation imitation rewards. P-SIL (Pixel Sinkhorn Imitation Learning) learns and compares latent representations of the encoded expert and agent behaviors; and P-SAC (Pixel Discriminator Actor Critic) augments the representations of the agents and the experts. The key component of this work is the RL encoder used in both P-SIL and P-SAC.

**Summary Of The Review:**

Imitation learning can be applied in tasks where a reward function is hard to specify or too sparse to be used in practice. The methods introduced in this work is useful for imitation learning from images directly. However, the learning from pixels can be ambiguous in certain cases since the actions cannot be fully described in a video.

A paper to appear in the Proceeding of Deep Reinforced Learning Workshop in NeurIPS 2021 has the same tile with this submission: Imitation Learning from Pixel Observations for Continuous Control (https://nips.cc/Conferences/2021/Schedule?showEvent=21848).

---

> ### Author Response · Authors · 2021-11-17
> **Rebuttal**
>
> Thank you for your time reviewing our paper and giving feedback! We are glad you considered our work useful for imitation learning from images directly. We would like to clarify some points of misunderstanding on the experimental details, and address the concerns raised below and in the paper.
>
> **It is not clear how to augment the data in practice:**
> All baselines leverage a DrQ-v2 backbone, which has a data augmentation strategy that consists of random shifts with padding and a random crop to restore the original image dimension, followed by bilinear interpolation. We clarified this further at the end of page 3 in the paper (in red).
>
> **It is not always possible to used the RL encoder for many tasks. This is the key issue of this work + the learning from pixels can be ambiguous in certain cases since the actions cannot be fully described in a video:**
> We learn the RL encoder online in the RL training loop, and leverage DrQ-v2 as the backbone, hence learning an encoder, a policy and a critic. Therefore, it is always possible to train an encoder in that way in the visual setting. Also, as per previous works, we leverage frame stacking which allows us to recover velocity and momentum information, which are useful to implicitly recover actions in the image setting, alleviating the ambiguity in actions.
>
> **The It is necessary to compare the proposed methods with other extentions of DAC and SIL:**
> We emphasize that all imitation baselines (besides BC), including DAC and SIL, P-DAC and P-SIL have a unified DrQ-v2 backbone, hence all learn an encoder in the same way in the RL loop. As a result, it is actually practical within our provided codebase to compare in a fair way to other potential extensions of DAC and SIL. If you have any suggestions of extensions of DAC and SIL, we would be interested in discussing this further. We also further clarified this at the end of page 3.
>
> **A paper to appear in the Proceeding of Deep Reinforced Learning Workshop in NeurIPS 2021...:**
> We note that this workshop has no proceedings, hence not violating the policy of ICLR 2022.

---

### Decision · Program_Chairs · 2022-01-20

**Decision:**

Reject

**Comment:**

Learning policies from video demonstrations alone without paired action data is a promising paradigm for scaling up Imitation Learning. As such the paper is well-motivated. Two approaches P-SIL and P-DAC train rewards for RL training, based on learning Sinkhorn distances between trajectory embeddings and an adversarial approach.  The reviews brought up lack of clarity in presentation and experimental results and ablation studies falling short of convincingly demonstrating value of distance functions used and other design tradeoffs. As such the paper does not meet the bar for acceptance at ICLR.